# Interplay between Cellular Autophagy and Hepatitis B Virus Replication: A Systematic Review

**DOI:** 10.3390/cells9092101

**Published:** 2020-09-15

**Authors:** Yong Lin, Zhenyu Zhao, Ailong Huang, Mengji Lu

**Affiliations:** 1Key Laboratory of Molecular Biology of Infectious Diseases (Chinese Ministry of Education), The Second Affiliated Hospital, Institute for Viral Hepatitis, Chongqing Medical University, Chongqing 400016, China; 2019111135@stu.cqmu.edu.cn (Z.Z.); ahuang@cqmu.edu.cn (A.H.); 2Institute of Virology, University Hospital Essen, University of Duisburg-Essen, 45122 Essen, Germany

**Keywords:** autophagy, hepatitis B virus, viral replication, HBx, HBs, autophagosome

## Abstract

Autophagy, a conserved process in which cells break down and destroy old, damaged, or abnormal proteins and other substances in the cytoplasm through lysosomal degradation, occurs via autophagosome formation and aids in the maintenance of intracellular homeostasis. Autophagy is closely associated with hepatitis B virus (HBV) replication and assembly. Currently, HBV infection is still one of the most serious public health issues worldwide. The unavailability of satisfactory therapeutic strategies for chronic HBV infection indicates an urgent need to elucidate the mechanisms underlying the pathogenesis of HBV infection. Increasing evidence has shown that HBV not only possesses the ability to induce incomplete autophagy but also evades autophagic degradation, indicating that HBV utilizes or hijacks the autophagy machinery for its own replication. Therefore, autophagy might be a crucial target pathway for controlling HBV infection. The definite molecular mechanisms underlying the association between cellular autophagy and HBV replication require further clarification. In this review, we have summarized and discussed the latest findings on the interplay between autophagy and HBV replication.

## 1. Introduction

Currently, hepatitis B virus (HBV) infection is one of the most serious public health issues worldwide [1,2,3]. Patients with HBV infection are at a greater risk of developing acute or chronic hepatitis, liver fibrosis or cirrhosis, and hepatocellular carcinoma (HCC) [4]. Although HBV vaccines have been widely used to reduce the incidence of HBV infection, current therapies using interferons and nucleoside/nucleotide analogues have limited efficacies to cure the disease [3,5].

HBV, a small DNA virus belonging to the *Hepadnaviridae* family, consists of a 3.2 kb genome made up of partially double-stranded DNA [6]. The HBV genome contains four overlapping open reading frames (ORFs) consisting of the S, C, P, and X regions, which encode three envelope proteins (small, SHBs; medium, MHBs; large, LHBs), the core and precore proteins (including HBeAg), polymerase, and the X protein, respectively. The mature HBV virion consists of nucleocapsids (NCs) and envelope proteins. The capsid/NC assembly occurs by the recruitment of the core proteins, pre-genomic RNA (pgRNA), and viral DNA polymerase, followed by the reverse transcription of the pgRNA and formation of the relaxed circular DNA [7]. Subsequently, the mature capsid/NCs are enveloped through their interactions with the LHBs, and they are released as mature virions [8]. Additionally, non-infectious subviral particles (SVPs) consisting of SHBs, MHBs, and LHBs are independently secreted from the host cell through the ER–Golgi export pathway.

Autophagy is a conserved “self-eating” process by which cytoplasmic materials including misfolded proteins, damaged organelles, and various invading pathogens are removed to maintain cellular homeostasis [9]. Autophagy occurs in four main steps: initiation, phagophore formation, phagophore expansion to form autophagosomes, and the fusion of autophagosomes with the lysosomes [10]. The initiation and formation of autophagosomes is known as early autophagy, while the final step is known as late autophagy. The initiation of phagophore formation is characterized by the activation of AMP-activated protein kinase (AMPK) signaling and/or the inhibition of mammalian target of rapamycin (mTOR) signaling, which causes the activation of the ULK1/ATG1 complex and its translocation into the ER. Subsequently, ULK1 activates the class III phosphoinositide 3-kinase complex I (PtdIns3K-C1, consisting of PIK3C3/VPS34, VPS15, BECN1/Beclin 1, and ATG14L) and recruited it to the endoplasmic reticulum (ER) membrane. This leads to the formation of the intermediate, omegasome, which is rich in phosphatidylinositol 3-phosphate (PtdIns3P). The omegasome formation is followed by the formation of phagophores, which expand and elongate with the help of two ubiquitination-like systems, the ATG5-12-16L conjugation complex and the LC3/ATG8 conjugation complex, to form double-membrane vacuoles called autophagosomes. Finally, the mature autophagosomes fuse with lysosomes and form autolysosomes, where the cellular waste is degraded.

An in-depth understanding of cellular autophagy and HBV replication has revealed the close association between these processes in the life cycle of HBV. Previous studies have shown that HBV proteins such as HBx or SHBs affect the autophagic flux either directly or indirectly [11,12,13]. Conversely, efficient HBV replication is dependent heavily on the autophagic machinery [14,15,16,17,18]. Previous reports have revealed that the early phase of autophagy is essential for HBV replication and envelopment. As a consequence, HBV infection may worsen HBV pathogenesis and lead to other related liver diseases including liver cancer [19,20]. Besides, previous studies reveal that autophagic degradation restricts or degrades viral particles and SVPs in a lysosome-dependent manner [14,18,21,22]. Thus, the definite molecular mechanisms underlying the association between cellular autophagy and HBV replication require further clarification. In this review, we have discussed and summarized the major findings in this regard.

## 2. HBV-Mediated Autophagic Responses

### 2.1. HBV Induces the Initiation of Autophagy

During starvation or stress conditions, the inhibition of mTOR signaling and/or activation of AMPK signaling induces the activation of the ATG1/ULK1 complex, which positively regulates the activity of the PIK3C3 complex in a phosphorylation-dependent manner [10].

mTOR signaling, one of the major oncogenic pathways, is a crucial regulator of a number of cellular processes including proliferation, differentiation, and survival [23]. Previous studies suggest that mTOR signaling plays an important role in suppressing HBV replication in hepatocytes [17,24,25,26]. Guo et al. demonstrated that the activation of the PI3K/protein kinase B (Akt)/mTOR signaling pathway decreases HBV DNA replication at the transcription level in different hepatoma cells [25]. Our previous study showed that the microRNA-99 family-mediated inhibition of the IGF-1R/PI3K/Akt/mTOR signaling pathway by the direct targeting of IGF-1R, Akt, and mTOR and induces a strong autophagic response, thereby promoting HBV replication [24]. Thus, the inhibition of mTOR signaling enhances HBV replication. Nevertheless, HBV proteins possess the ability to activate mTOR signaling. The HBV-associated proteins, HBx and pre-S, activate mTOR signaling during HBV infection (Figure 1). This process is closely linked to the pathogenesis of HBV-associated HCC [27,28,29]. Yen et al. reported that the HBx-activated mTOR signaling proceeds via IκB kinase β (IKKβ) and increases cell proliferation and vascular endothelial growth factor (VEGF) production during HCC development [28]. HBx also induces anti-apoptotic proteins and promotes cell cycle progression in hepatocytes in vivo by activating the mTOR signaling pathway [30]. Yang et al. found that HBV wild-type or mutant pre-S2 proteins enhance VEGF-A expression and activate mTOR signaling in ground glass hepatocytes (GGHs) of chronic HBV carriers [31]. Moreover, the HBx protein activates Akt and reduces HBV replication by reducing the activity of the transcription factor, hepatocyte nuclear factor 4alpha (HNF4α), while promoting the survival of primary hepatocytes [32]. Another study demonstrated that activated mTOR signaling causes the feedback inhibition of HBsAg synthesis in dysplastic GGHs and HCC tissues [26]. Moreover, the pre-S1 protein-induced activation of mTOR results in the recruitment of the transcription factor, Yin Yang 1 (YY1), and its co-repressor, histone deacetylase 1 (HDAC1) complex, which leads to the feedback inhibition of pre-S1 transcription. HBx-mediated activation of mTOR signaling is supposed to result in the inhibition of hepatocyte apoptosis, thereby facilitating persistent HBV replication.

HBV infection activates the key energy sensor AMPK (Table 1), which regulates cellular metabolism and maintains energy homeostasis [33,34]. AMPK is also involved in the regulation of autophagy. During starvation, AMPK promotes autophagy induction by directly activating ULK1 through the phosphorylation of Ser 317 and Ser 777 [33]. During normal conditions, this is prevented by high mTOR activity, which prevents ULK1 activation by phosphorylating Ser 757 of ULK1, thereby disrupting the interaction between ULK1 and AMPK. This coordinated phosphorylation is important for the activation of ULK1 during the regulation of autophagy induction. Additionally, the starvation-induced reactive oxygen species (ROS) generated in the mitochondria lead to the autophagy-mediated activation of AMPK signaling [35]. Tang et al. reported that HBx increases the levels of autophagy-associated proteins during starvation by upregulating BECN1 expression, indicating that HBx contributes to the growth of HBV-infected hepatocytes under nutrient-deficient conditions [11]. Moreover, our recent work revealed that the inhibition of AMPK signaling at high glucose concentrations leads to the inhibition of AMPK/mTOR-ULK1-induced autophagy, thereby reducing HBV replication in hepatoma cells and primary hepatocytes [36]. Collectively, AMPK activation-mediated autophagy regulates HBV replication.

Interestingly, Bagga et al. showed that HBx activates both the mTOR and AMPK signaling pathways in cultured primary rat hepatocytes, revealing the requirement of a balance between these pathways for the promotion of persistent HBV replication [34]. Thus, a study of the factors associated with the mTOR and AMPK signaling pathways might reveal potential therapeutic targets for the inhibition of HBV replication and prevention of HBV-associated liver cancer.

### 2.2. HBV Promotes Phagophore Formation

Upon activation, PtdIns3K forms PtdIns3P in certain ER domains. It plays a vital role in omegasome formation (in mammals) and PtdIns3K-C1 recruitment, and it drives the lipid changes necessary for phagophore formation [10]. Numerous studies have reported that HBV induces partial autophagy through PtdIns3K-C1 activation and phagophore formation, and it facilitates its own replication through the actions of HBx in hepatocytes [13,38]. First, HBx directly binds to PtdIns3K, enhances the enzymatic activity of PtdIns3K, and induces autophagy for viral DNA replication in Huh7.5 cells [38]. PtdIns3K activation is crucial for HBx-induced autophagosome formation [11,38,39]. Moreover, HBx upregulates BECN1 activity and induces autophagosome formation either directly or indirectly. BECN1 interacts with different activators including ATG14/ATG14L, UVRAG/Bax-interacting factor-1 (Bif-1), and RUN domain and cysteine-rich domain containing (Rubicon), and it regulates the different phases of autophagy. Tang et al. found that BECN1 expression was increased in HBV-infected cancerous liver tissues and HBx-overexpressing hepatoma cell lines [11]. During nutrient-deficient conditions, HBx directly transactivates BECN1 promoter activity and upregulates its expression to promote autophagy. However, Wang et al. reported that HBx activates BECN1-mediated autophagy through the C-myc/miR-192-3p/X-linked inhibitor of apoptosis protein (XIAP)/NF-κB axis in HepG2.2.15 or Huh7 cells. They also found that autophagy is essential for HBV replication both in vitro and in vivo [40]. In addition, HBx also induces starvation-induced autophagy by increasing the activity of death-associated protein kinase (DAPK) in Chang cells in a BECN1-associated pathway [12]. However, another study by Zhong et al. showed that HBx does not alter the expression levels of PIK3C3 and BECN1 but dissociates BECN1 and Bcl-2 in HepG2 cells [39]. They further found that HBx leads to the generation of ROS, activation of c-Jun N-terminal kinases (JNK) signaling, and it releases BECN1 from its association with Bcl-2 to form a complex with PIK3C3-C1, thereby activating the autophagic flux. Thus, the regulation of autophagy induction, which is critical for HBV replication, occurs via the ROS/JNK signaling pathway. Collectively, HBV induces phagophore formation by activating and recruiting PtdIns3K-C1 and driving the lipid composition changes necessary for phagophore formation. 

### 2.3. HBV Activates Phagophore Expansion and Forms Autophagosomes

Two ubiquitination-like systems, the ATG5-12-16L1 and LC3 conjugation complexes, are activated to expand and elongate the phagophore membranes [9]. Following binding of the ATG5-12-16L1 complex to the phagophore, and LC3 lipidation, the phagophore elongates and engulfs the substrates in the cytoplasm, ultimately leading to the formation of a complete autophagosome.

ER stress induces unfolded protein response (UPR) through activating protein kinase RNA-like ER kinase (PERK), transcription factor 6 (ATF6), and inositol requiring protein-1α (IRE1α) signaling, which are linked with autophagy initiation [42]. First, SHBs not only activate IRE1α/XBP1/BECN1 axis-mediated autophagy induction in hepatoma cells, but also activate PERK and ATF6 signaling, in which the downstream molecules eIF2α and GRP78/94 interact with the autophagy-associated proteins ATG5, ATG12, and/or ATG16L to induce UPR-linked autophagy [13,37]. SHBs enhance the autophagic process without promoting lysosomal protein degradation, indicating that SHB-induced UPR signaling is critical for phagophore elongation and autophagosome maturation. Second, HBV subverts the formation of the autophagy elongation complex ATG5-12-16L1 and proceeds to NC assembly/envelope maturation. Although ATG8/LC3 lipidation is dispensable for this process, the LC3 complex interacts with SHBs [15]. This implies that autophagosomes provide a physical scaffold for the envelopment of HBV and serve as a source of membranes for this process [42]. However, another study showed that HBV activates the ER-associated degradation (ERAD) pathway in Huh7 or HepG2 cells, which relieves ER stress during UPR and induces the degradation of HBs by ER degradation-enhancing mannosidase-like proteins (EDEMs) [22,43]. Thus, ER stress caused by the accumulation of SHBs might induce autophagy and ERAD at the same time. However, the existence of a balance between them controls the level of viral particles in infected cells.

### 2.4. HBV Interferes with Autophagic Degradation

Previous studies have suggested that HBV induces autophagosome formation but inhibits the complete autophagic process. HBx-induced autophagosome formation occurs due to the reduced degradation of both LC3 and SQSTM1/p62, indicating that HBV-induced autophagosome formation is enhanced without the promotion of autophagic degradation by lysosomes [16,41]. 

HBV assembly is closely linked to the autophagic process. However, the different stages of autophagy impose different effects on HBV replication. The autophagic degradation process primarily depends on the fusion between the autophagosome and lysosome and the lysosomal function [9]. HBV infection blocks the fusion between the autophagosome and lysosome in hepatocytes. Our group found that a proportion of SVPs and NC/virion-associated HBV DNA was degraded following the fusion of autophagosomes and lysosomes in primary hepatocytes and different hepatoma cells [21,44]. Furthermore, in concordance with previous reports, our and other laboratories found that HBV blocks the fusion of autophagosomes with lysosomes and prevents autophagic degradation by decreasing the expression of Rab7 and SNAP29 [18,21,41]. These findings present novel mechanisms by which HBV induces incomplete autophagy to escape autophagic degradation. Another mechanism by which HBV evades autophagic degradation is by impairing lysosomal and autolysosomal acidification. Liu et al. reported that HBx impairs lysosome maturation by inhibiting its acidification without disturbing the autophagosome–lysosome fusion in Huh7 cells, which is beneficial for HBV replication [16]. They further found that HBx interacts with vacuolar-type H^+^-ATPase (V-ATPase) and impairs lysosomal acidification, leading to reduced lysosomal degradation and the accumulation of immature lysosomes. In addition, a study by Xiong’s group reported that HBV induces incomplete autophagy by impairing lysosomal acidification [45]. They further revealed that epigallocatechin-3-gallate (EGCG) treatment efficiently inhibits HBV replication by preventing HBV-induced incomplete autophagy via the enhancement of lysosomal acidification in HepG2 cells. Thus, this study suggests that drugs enhancing autophagic degradation can be used as adjuvants in anti-HBV agents.

Collectively, these findings suggest that HBV subverts autophagy not only by inducing autophagosome formation to aid in its own replication but also by blocking lysosomal degradation to avoid its degradation.

## 3. Autophagy-Mediated HBV Replication and Assembly

### 3.1. HBV Manipulates Autophagic Components for Its Replication

Various studies have demonstrated the essential role of autophagy induction in HBV replication [13,18,24,38,40,44,46]. Conversely, autophagy suppression prevents HBV replication both in vitro and in vivo. 

Autophagy plays a positive role in HBV replication and assembly in hepatoma cells with stable HBV replication, cells transiently transfected with replication-competent HBV genomes, and HBV virion-infected cells (Figure 2). Sir et al. reported a reduction in HBV DNA replication when cells were treated with the PtdIns3K inhibitor, 3-methyladenine (3-MA), or with a specific siRNA targeting the *PIK3C3* or *ATG7* genes. Thus, the suppression of PtdIns3K activity or inhibition of *ATG7* expression results in reduced HBV replication [23]. An increasing number of studies have reported that certain drugs, including rapamycin, cisplatin, and dexamethasone, induce cellular autophagy and enhance HBV replication [47,48,49]. Chen et al. revealed that cisplatin induces autophagy by activating the ROS/JNK signaling pathway and inhibiting the Akt/mTOR signaling pathway in both HBV-replicating cells and an HBV-transgenic mouse model [49]. Moreover, our previous study found that the non-coding microRNA-99 (miR-99) family including miR-99a, miR-99b, and miR-100 promotes HBV replication post-transcriptionally by targeting IGF-1R, Akt, and mTOR through IGF-1R/PI3K/Akt/mTOR/ULK1 signaling-induced autophagy [24]. Another study by Yang et al. reported that microRNA-141 inhibits autophagy by targeting sirt1 and reducing HBV replication [50]. Under glucose starvation conditions, HBV replication is greatly enhanced and regulated by AMPK via Akt/mTOR signaling-mediated cellular autophagy [36]. 

Autophagy is not only essential for HBV replication in cell cultures, but it is also important for its replication in vivo. Tian et al. used HBV transgenic mice with liver-specific knockout of the *ATG5* gene to demonstrate a significant reduction in the HBV DNA level in the sera. They also found that HBV DNA replication was almost eliminated in the mouse liver [46]. Moreover, our recent study revealed that glucosamine application induces the formation of a large number of autophagosomes and greatly enhances HBV replication by suppressing its autophagic degradation and by inhibiting mTOR signaling in the liver tissue of HBV hydrodynamic injection mouse model [44]. These evidences suggest that enhanced autophagy promotes HBV replication in vivo.

Although the close association between HBV replication and autophagy has been identified, the exact stages of the HBV life cycle affected by autophagy are not completely understood. In 2010, Sir et al. revealed that the suppression of autophagy by 3-MA or a specific siRNA targeting *ATG7* had no significant effect on the level of HBV precore RNA but slightly reduced the level of core RNA (pgRNA) packaged in core particles in Huh7.5 cells. This indicates that autophagy plays an important role in HBV DNA replication and affects the level of HBV core RNA packaged in core particles [38]. Additionally, our previous studies confirmed that autophagy has a marginal effect on HBV transcription and HBeAg production [13,18,24,36,38]. Recent studies have revealed that HBV-induced ROS generation facilitates HBV replication and capsid assembly via ROS/JNK signaling-induced autophagosome formation [20]. Thus, autophagy enhances HBV replication primarily at the viral DNA replication stage. The ATG5-12-16L elongation complex provides a physical scaffold for HBV replication and maturation. To study HBV core/capsid maturation in the absence of ongoing viral replication, Döring et al. used two special expression constructs pCore and pDPAF.Core without HBV RT and related envelope protein (Env) ORFs. Their study revealed that the proper formation and release of HBV naked particles requires the function of the RAB33B and ATG5-12-16L1 elongation complexes in the absence of a complete and functional autophagy pathway [51]. Their recent studies have revealed that the ATG5-12-16L1 complex induces the translation of the HBV core protein and mediates the trafficking of the core protein to NC assembly/envelope generation sites [15]. The silencing of *ATG5*, *ATG12*, and *ATG16L1* suppresses core protein translation and hampers proper core protein trafficking to NC assembly/envelope generation sites, resulting in the impaired association of core proteins/NCs with envelope membranes. HBV accesses the ATG5-12-16L complex via a direct interaction of its core protein with the intrinsically disordered region of ATG12. Collectively, HBV manipulates the autophagy machinery for its own replication. 

### 3.2. HBV Utilizes Autophagic Elements for its Envelopment

Mature NCs are enveloped by HBs proteins and released as progeny virions from the cell through exocytosis [52]. HBV exploits the elements of the autophagy system for its envelopment. Li et al. reported that the autophagy machinery was required for HBV envelopment but not for efficient viral budding [13]. They found that the SHBs partially co-localize and interact with the autophagic marker, LC3, thereby proving that autophagosomes contribute to viral envelope formation. Another study demonstrated that HBVs utilize the ATG5-12-16L1 elongation complex as a physical scaffold for viral replication and envelopment [15]. ATG12 has been confirmed to be associated with the HBV core. However, pulldown experiments did not reveal any direct interaction between the HBV core and LC3B, implying that the co-localization of the HBV core with LC3B is mediated through ATG12 tethered to the phagophore. Interestingly, the inhibition of LC3 lipidation increases the amounts of extracellular virions, suggesting that HBV induces early, non-destructive autophagy without culminating in the destructive autolysosomes. Additionally, autophagosome membranes serve as a source of membranes for viral envelope formation. Although various studies have confirmed that HBs are located in the autophagosome membranes, they have not clarified the source of these membranes. Compelling evidence suggest that the ER (where a large amount of HBs are located) serves as the primary source of autophagosome biogenesis [53]. An increasing number of studies have reported that the ER-derived COPII-coated vesicles, which bud at the ER exit site (ERES), are redirected away from the Golgi and toward the nascent phagophore, thereby contributing to autophagosome formation in both yeast and mammalian cells [54,55,56]. A recent study by Zeyen et al. has found that the HBV envelope selectively encounters the SEC24A/SEC23B complex in COPII vesicles through a direct interaction involving the N-terminal of SEC24A and a di-arginine motif of its S domain, indicating that viral envelope coating in COPII vesicles serves an important source of autophagy mediated-HBV envelope formation [57]. Therefore, autophagosomes and other autophagic compartments provide a physical scaffold for HBV envelopment and serve as a source of viral envelope membranes.

### 3.3. Late Autophagy Degrades HBV NCs/Virions and SVPs

Since autophagy is a conservative catabolic process, we assume that a significant part of HBV NCs/virions and other components are eliminated by autophagic degradation.

Numerous studies have shown that the late stage of autophagy is responsible for degrading old, damaged, and abnormal proteins, including HBV NCs/virions and SVPs [13,18,38]. Xie et al. observed that HBV replication was restricted by the activation of the PRKAA/AMPK-induced autophagic degradation process [14]. They further revealed the definite mechanism by which PRKAA/AMPK activation promoted the autolysosome-dependent degradation of cytoplasmic waste through the enhancement of cellular ATP levels, resulting in the degradation of the HBVs in autophagic vacuoles. Moreover, an interesting study by Zhong et al. showed that EGCG, a major polyphenol in green tea, prevents HBV-induced incomplete autophagy and induces complete autophagy to degrade HBV by enhancing lysosomal acidification in a process similar to starvation [45]. In our previous work, we found that HBV NCs/virions and SVPs were partly subjected to RAB7 complex-mediated lysosomal degradation [18]. Similarly, our recent findings have shown that autophagic degradation, which is controlled by the SNAP29 complex, critically determines the production of HBV NCs/virions and SVPs [21]. Moreover, we found that *RAB7* and *SNAP29* silencing with specific siRNAs led to the increased accumulation of autophagosomes and intracellular HBV NCs/virions and HBs proteins. Thus, SNAP29 and RAB7 act synergistically and regulate the autophagic degradation of HBV NCs/virions and SVPs [21]. These findings demonstrate the role of RAB7 and SNAP29 in mediating the fusion of autophagosomes with lysosomes and HBV production. Our recent study found that glucosamine, a natural compound, effectively promotes HBV replication and gene expression in vitro and in vivo by interfering with lysosomal acidification and degradation, and by promoting autophagy initiation through feedback inhibition of mTOR signaling [44]. Additionally, the activation of PRKAA (a catalytic subunit of AMPK)/AMPK by AICAR, an AMPK agonist, reduces the production of HBV particles by promoting their autophagic degradation [14]. Conversely, compound C, an AMPK inhibitor, inflicts opposite effects on HBV production. They further found that active PRKAA/AMPK promotes the autolysosome-dependent degradation of HBV through the stimulation of cellular ATP levels, leading to the degradation of HBV particles. These studies indicate that the late stage of autophagy restricts HBV replication owing to the partial autophagic degradation of HBV NCs/virions and SVPs.

Additionally, Lazar et al. reported the involvement of the endoplasmic reticulum-associated degradation (ERAD) pathway in the degradation of HBs proteins through the increased expression of ER degradation-enhancing mannosidase-like proteins (EDEMs). This leads to a reduction in the amount of intracellular HBs and relief of ER stress [43]. Furthermore, we found that suppressor of lin-12-like 1 (SEL1L), a vital component of the ERAD pathway, exerts an anti-HBV effect through the ERAD and alternative ER quality control (ERQC)-autophagy pathways [22].

Since autophagic degradation partially eliminates HBV NCs/virions and SVPs, HBV develops strategies to evade this process. Previous studies have found that HBx partly evades autophagic degradation by impairing lysosomal acidification, which is achieved by inhibiting V-ATPase activity or by decreasing RAB7 and SNAP29 expression by blocking the fusion of the autophagosome with the lysosome [16,18,21,41]. Efficient HBV replication and assembly requires the early stage of the autophagic process but not the late stage that involves lysosomal degradation [44]. 

### 3.4. HBV Evades Cell Death via HBX-Induced Autophagy for Persistent Infection

HBV adopts certain strategies to manipulate apoptotic signaling pathways and utilize them for replication, resulting in the persistent infection of host cells. Numerous studies suggest that HBx is an oncogene involved in the development of HCC. Liu et al. found that HBx significantly increased methionine adenosyltransferase 2A (MAT2A) expression by activating the promoter of MAT2A through the NF-κB and CREB signaling pathways in hepatoma cells [58]. Thus, the close relationship between HBx and MAT2A inhibits apoptosis in hepatoma cells and enhances HCC development. Moreover, HBx interferes with the association between the death ligand, tumor necrosis factor superfamily member 10 (TNFSF10/TRAIL), and its receptor, TNF receptor superfamily member 10b (TNFRSF10B/DR5), which participate in the immune surveillance of virus-infected or -transformed cells [59]. Thus, HBx plays an important role in disrupting the binding of TNFSF10 to TNFRSF10B, and it permits cell survival via the recruitment of TNFRSF10B to LC3B for autophagic degradation. Additionally, HBx reduces cell death and promotes cell survival during starvation. Mao et al. found that HBx inhibits caspase-3 activity and blocks the mitochondrial apoptotic pathway induced by starvation [60]. Moreover, HBx promotes the survival of HBV-infected hepatocytes during nutrient-deprived conditions by inhibiting apoptosis and activating autophagy. These studies suggest that HBx-induced autophagy is a cell survival factor facilitating persistent HBV infection.

### 3.5. Autophagy Regulates HBV-Related Immune Responses

An increasing amount of evidence has revealed the intrinsic interactions between autophagy and HBV-related immune responses. Currently, various evidences reveal that autophagy has multiple effects on immunity [61]. Additionally, autophagy plays an important role in HBV-mediated innate or adaptive immune responses.

Autophagy is a potent regulator of innate immunity-induced inflammation. As an essential part of cellular defense, autophagy is naturally involved in innate immunity. First, autophagy impairs the interferon signaling pathways, thereby benefiting HBV replication. Kunanopparat et al. reported that *ATG12* knockdown by shRNA in HepG2.2.15 cells decreased the amount of HBV and increased the expression of interferon-alpha (IFN-α), interferon-beta (IFN-β), and interferon-inducible (IFI) genes, suggesting that *ATG12* plays a crucial role in HBV replication by downregulating the IFN pathway [62]. Luo et al. reported that autophagy inhibition abrogates the HBx-induced activation of NF-κB and the production of pro-inflammatory cytokines including IL-6, IL-8, and CXCL2 [63]. These findings suggest that autophagy promotes HBx-induced inflammatory responses. Wang J. et al. found that the induction of autophagy by HBV infection was important for the maintenance of HBV replication through the miR-192-3p-XIAP axis by NF-κB signaling both in vitro and in vivo [40]. This suggests that HBV infection modulates the miR-192-3p-mediated autophagy response and induces HBV evasion. Second, Shin et al. found that HBx possesses dual functions. It induces autophagy and promotes the association of TNFRSF10B with LC3B, an autophagy receptor-like molecule involved in TNFRSF10B-mediated degradation [59]. Thus, HBx facilitates HBV evasion from TNFSF10-mediated antiviral immunity and contributes to chronic HBV infection. Moreover, Hu S. et al. reported that the HBV core and E proteins inhibit the extracellular release of neutrophils by decreasing ROS production and inhibiting autophagy, thereby evading elimination and resulting in the establishment of chronic HBV infection [64]. Additionally, our colleagues found that the level of Rubicon, a BECN1-binding partner that negatively regulates the initiation of autophagy, was elevated in the peripheral blood mononuclear cells, sera, and liver tissues of patients with HBV infection compared to healthy individuals [65,66]. Further study revealed that Rubicon overexpression significantly promoted HBV replication, which is modulated by the binding of Rubicon to NF-κB essential modulator (NEMO), leading to the inhibition of type-I IFN production [65]. However, it is unclear if Rubicon utilizes the Rubicon-mediated autophagy pathway to promote HBV replication [67]. Moreover, the autophagy-mediated suppression of innate immunity is highly likely to enhance HBV replication.

Autophagy is essential for HBV-specific adaptive immune responses. Ma et al. demonstrated the ability of a lentivector-targeting engineered dendritic cell (DC) to induce robust T-cell responses and elicit effective anti-HBV effect in HBV transgenic mice [68,69]. They further revealed that the upregulation of the autophagy of T cells aids in the eradication of HBV by promoting the survival and proliferation of activated CD8^+^ T cells. However, Cheng LS et al. reported that high-mobility group box-1 (HMGB1)-induced autophagy was essential for maintaining cell survival and the functional stability of Treg cells during chronic HBV infection [70]. These studies offer different perspectives for the development of therapeutic strategies that activate specific anti-HBV immune responses by specifically modulating the autophagy process of T cells or Tregs. Collectively, HBV subverts or hijacks the autophagy process to evade immune clearance.

## 4. Autophagy-Based Anti-HBV Strategies

Various studies have broadened our knowledge on the interplay between autophagy and HBV replication. In particular, numerous amounts of evidence suggest that the modulation of autophagy is a potential therapeutic strategy for controlling HBV infection. The inhibition of autophagy initiation and enhancement of autophagic degradation are the two main strategies for the elimination of HBV [20]. Moreover, novel compounds or drugs that inhibit autophagy initiation display good anti-HBV effects. NTI-007, a novel synthesized compound targeting the HBx-BECN1 pathway, shows effective anti-HBV potential in vitro. Given the high efficacy of NTI-007 in triggering autophagy, the NTCP-mediated autophagic pathway serves as a promising strategy for HBV therapy [71]. On the contrary, drugs that enhance autophagic degradation also demonstrate potential therapeutic applications. Since ROS-induced AMPK activation promotes the autophagic degradation of HBV particles, AMPK agonists (such as AICAR) are utilized as potential anti-HBV agents [14]. Similarly, EGCG treatment efficiently promotes the autophagic degradation of HBV particles, thus inhibiting HBV replication by enhancing lysosomal acidification [45]. Additional therapeutic strategies for study in a chronic HBV infectious mouse model include the clearing of the HBV-infected hepatocytes and breaking of HBV tolerance by inducing anti-HBV CD8^+^ effector T cells [72]. Recently, lentivector-targeting engineered DCs have been safely administered to patients. Ma et al. constructed a DC directed against a lentivector (LV) (LVDC-UbHBcAg-LIGHT) for use as a potential vaccine to induce anti-HBV immune responses. They found that the vaccine elicited effective anti-HBV effects in HBV transgenic mice [68,69]. Further study revealed that the increased autophagy of T cells primarily contributed to the elimination of HBVs. Collectively, these autophagy-related drugs provide new therapeutic options for the treatment of chronic HBV infections.

## 5. Conclusions and Perspective

In summary, we have provided a detailed description of the recent advances in the studies on the interplay between cellular autophagy and HBV replication. Autophagy plays a crucial role in the HBV life cycle. It aids in viral NC assembly/stability, envelopment, and degradation. HBVs hijack the autophagic process, evade cell death, and inhibit HBV-related immune responses, benefiting their own persistence. In particular, an increasing number of studies have shown that HBV induces incomplete autophagy and evades autophagic degradation for its own efficient replication and survival [19,73]. HBV-associated proteins, including HBx and HBs, are crucial factors involved in HBV-mediated autophagy [13,16,38,40]. HBV utilizes the components of early and non-degrading autophagy as assembly scaffolds [15]. Recently, numerous studies have revealed that HBV NCs/virions or SVPs are partially degraded during the late phase of autophagy [14,18,21]. However, HBV adopts certain strategies to evade this degradation [14,18,21]. Currently, it is well accepted that SVPs are mainly secreted during the constitutive secretory pathway of the ER/ERGIC/Golgi axis [57,74]. However, the budding of virions is closely associated with the multi-vesicular body (MVB) pathway [75,76]. To date, it is unclear if HBV NCs/virions or SVPs are released during the autophagic secretion pathway [77,78], thereby warranting further investigation. Although the relationship between autophagy and HBV infection has been gradually revealed, numerous unresolved issues exist.

HBV may exploit autophagy for its own persistence in patients with HBV infection. Up to date, the molecular mechanisms underlying the association between cellular autophagy and HBV replication were mainly investigated in vitro or in a mouse model. Yet, there have been limited studies on the clinical relevance of autophagy and HBV replication in humans. The induction of autophagosomes by HBV was observed in the liver of HBV-infected patients but not in the liver of non-HBV-infected patients [16,38]. Nevertheless, the association of HBV replication and assembly with cellular autophagy may explain some clinical findings. A clinically relevant observation is that the circulating HBV particles secreted carry liver-specific microRNAs such as miR-122 [79,80,81]. Winther et al. found that there is a positive correlation between the plasma levels of 11 identified microRNAs and HBsAg over time from children with CHB [82]. A possible reason for this observation is that HBV particles are generated by the autophagic process and exported in the same way as secreted microRNAs packaged in extracellular vesicles [15,51,75]. In addition, HBV variants carrying mutation(s) in the preS/S genomic region contribute to the pathogenesis of patients with occult HBV infection, as well as fulminant hepatitis and fibrosing cholestatic hepatitis [83]. The definite molecular mechanisms of liver pathology caused by HBV preS/S mutants have not been fully defined yet. PreS/S variants may cause the accumulation of mutated HBsAg within the ER of hepatocytes [83,84,85]. Subsequently, the HBV mutants with defected HBsAg secretion may cause ER stress, which in turn enhances HBV replication and gene expression, inducing acute exacerbation or even liver failure. It is of great interest to control ER stress, which plays an important role in autophagy, HBV replication, and liver pathology. Experimentally, we previously demonstrated that the coexistence of two different HBV variants with secretion-defective HBsAg significantly induced specific host immune responses and enhanced immune-mediated liver damage in a mouse model [86]. However, the link of ER stress, HBV replication, and liver damage needs to be verified in future studies in patients.

During chronic HBV infection, HBV-induced autophagy worsens the condition of the infected liver and contributes to the pathogenesis of HBV-associated HCC [27,28,29]. Existing research suggests that the autophagy process is a potential therapeutic target for controlling HBV infection and HBV-related liver diseases [87,88]. Therefore, the use of a combination of autophagy-related drugs and anti-HBV drugs represents a potential therapeutic approach for clinical use.

## Figures and Tables

**Figure 1 cells-09-02101-f001:**
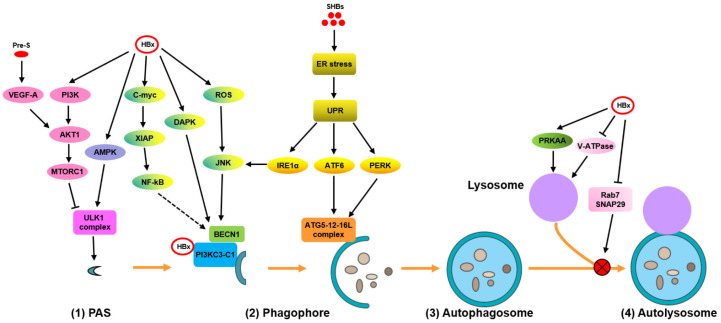
Hepatitis B virus (HBV) infection modulates the different phases of autophagy and autophagy-mediated host responses. (**1**) The initiation of autophagy. Autophagy is initiated by the activation of the ULK1 complex through mammalian target of rapamycin (mTOR) signaling inhibition and/or AMP-activated protein kinase (AMPK) signaling activation. (**2**) Phagophore formation: HBx induces phosphoinositide 3-kinase complex I (PtdIns3K-C1)-mediated autophagy by targeting the PtdIns3K, death-associated protein kinase (DAPK), reactive oxygen species/c-Jun N-terminal kinases (ROS/JNK), and c-myc-XIAP-NF-κB pathways either directly or indirectly. (**3**) Phagophore expansion and form autophagosomes: HBV subverts the autophagy elongation complex ATG5-12-16L1 for NC assembly/envelope maturation. SHBs trigger endoplasmic reticulum (ER) stress-induced UPR and activate IRE1α and ATF6 signaling to induce UPR-related autophagy. (**4**) Interference of HBV with autophagic degradation to form autolysosomes: HBV delays the fusion of autophagosomes with lysosomes by decreasing RAB7 or SNAP29 expression or by impairing lysosomal and autolysosomal acidification through the inhibition of V-ATPase activity. NC, nucleocapsid; PAS, pre-autophagosomal structure; ULK1, unc-51 like autophagy activating kinase 1; UPR, unfolded protein response.

**Figure 2 cells-09-02101-f002:**
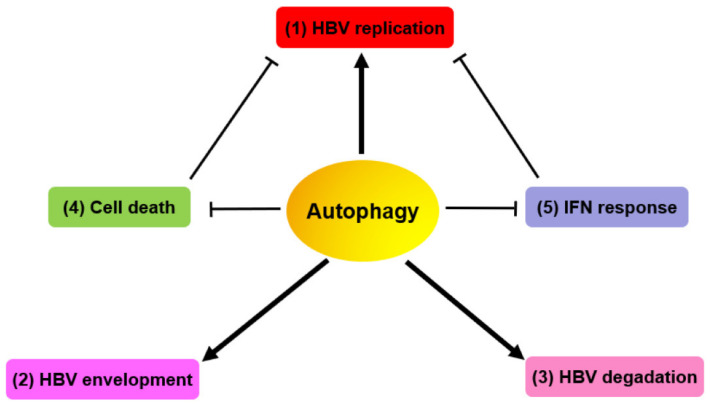
Autophagy and autophagy-mediated host responses regulate HBV replication and assembly. (**1**) HBV manipulates the autophagic components for its own replication. (**2**) HBV utilizes autophagic elements for envelope formation. (**3**) HBV NCs/virions and SVPs are degraded during the late stage of autophagy. (**4**) HBV evades cell death via HBx-induced autophagy for persistent infection. (**5**) Autophagy impairs the interferon signaling pathways, thereby benefiting HBV replication.

**Table 1 cells-09-02101-t001:** Mechanisms of HBV-mediated autophagic responses.

HBV Proteins.	Models	Mechanisms	Effects	References
HBx	HepG2.2.15 and HepAD38 cells	Activate AMPK by ROS accumulation	Induce autophagy initiation (?)	[14]
			Promote autophagic degradation	
HBx	Primary rat hepatocytes; HepG2.2.15 cells and primary hepatocytes	Activate AMPK signaling	Induce autophagy initiation	[34,36]
HBx	L02, Chang, HepG2, and BEL-7404 cells	Directly transactivates BECN1 promoter activity and upregulates its expression during starvation	Promote phagophore formation	[11]
HBx	Chang cells	Increase the activity of DAPK in a BECN1-associated pathway	Promote phagophore formation	[12]
SHBs	Huh7 cells	Activate IRE1α/XBP1/BECN1 axis	Promote phagophore formation	[13,37]
HBx	Huh7.5 cells	Directly bind to PtdIns3K and enhance its enzymatic activity	Promote phagophore formation	[38]
HBx	HepG2 cells	Dissociate BECN1 and Bcl-2 via the ROS/JNK signaling pathway	Promote phagophore formation	[39]
HBx	HepG2.2.15 and Huh7 cells; primary human hepatocytes; hydrodynamic-based HBV mouse model	Activate BECN1-mediated autophagy through C-myc/miR-192-3p/XIAP/NF-κB axis	Promote phagophore formation	[40]
SHBs	Huh7 cells	Activate PERK/eIF2α and ATF6/GRP78/94 signaling to enhance their interaction with the autophagy-associated proteins ATG5, ATG12, and/or ATG16L	Activate phagophore expansion and form autophagosomes	[13,37]
HBx	Huh7 cells	Impair lysosome maturation by inhibiting its acidification	Interfere with autophagic degradation	[16]
HBV	HepG2.2.15 and Huh7 cells	Block the fusion of autophagosomes with lysosomes by decreasing the expression of Rab7 and SNAP29	Interfere with autophagic degradation	[18,21,41]

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
