# Peer review of "Interplay between Cellular Autophagy and Hepatitis B Virus Replication: A Systematic Review"

_cells, 2020, doi:10.3390/cells9092101_

Round 1
Reviewer 1 Report
Lin et al. overviewed the associtions between HBV pathogenesis and autophagy. The authors summarized how HBV mediates host autophagy and vice versa. The review is well-written and comphrehensive. There are some comments to make it more reader-friendly.
Comments
- This review is conprehansive, but a little bit hard to understand which mechanisms of autophagy are affected by which part of HBV replication. Therefore, that would be helpful if the authors summarize reported mechanism in each step in Table format. Information on which mechanism was reported in vitro or in vivo experiment would be also helpful to understand.
- It is unclear that autophagy in which cells in the liver is mediated by HBV in sections. Almost all should be in hepatocyte. Please clarify the main cell in each mechanism.
- Clinical relevance will make the review more attractive. In the mechanisms the authors summarized, which one is actually observed in human?
- This reviewer suggests a couple of minor corrections of typos
line 47 in -> in,
line 260 evelop -> envelope?.
Reviewer 2 Report
This syatematic review manuscript was well-written and informative.
Author Response
We are thankful for the encouraging comment.
Reviewer 3 Report
This paper is very interesting with an extensive bibliography, well written and with adequate conclusions.
Author Response

(The authors gave the same response as above.)
